# *PAK1* copy number in breast cancer— Associations with proliferation and molecular subtypes

**Anette H. Skjervold**[1]*, **Marit Valla**[1,2], **Borgny Ytterhus**[1], **Anna M. Bofin**[1]

**1** Department of Clinical and Molecular Medicine, Faculty of Medicine and Health Sciences, Norwegian University of Science and Technology, Trondheim, Norway, **2** Department of Pathology, St. Olav's Hospital, Trondheim, Norway

* anette.skjervold@ntnu.no

## Abstract

**Data Availability Statement:** The datasets generated in the current study are not publicly available due to ethical and legal restrictions imposed by General Data Protection Regulations

## Introduction

P21-activated kinase 1 (*PAK1)* is known to be overexpressed in several human tumour types, including breast cancer (BC). It is located on chromosome 11 (11q13.5-q14.1) and plays a significant role in proliferation in BC. In this study we aimed to assess *PAK1* gene copy number (CN) in primary breast tumours and their corresponding lymph node metastases, and associations between *PAK1* CN and proliferation status, molecular subtype, and prognosis. In addition, we aimed to study associations between CNs of *PAK1* and *CCND1*. Both genes are located on the long arm of chromosome 11 (11q13).

## Methods

Fluorescence *in situ* hybridization for *PAK1* and Chromosome enumeration probe (CEP)11 were used on tissue microarray sections from a series of 512 BC cases. Copy numbers were estimated by counting the number of fluorescent signals for *PAK1* and CEP11 in 20 tumour cell nuclei. Pearson's $x^2$ test was performed to assess associations between *PAK1* CN and tumour features, and between *PAK1* and *CCND1* CNs. Cumulative risk of death from BC and hazard ratios were estimated in analysis of prognosis.

## Results

We found mean *PAK1* CN $\geq$4<6 in 26 (5.1%) tumours, and CN $\geq$ 6 in 22 (4.3%) tumours. The proportion of cases with copy number increase (mean CN $\geq$4) was highest among HER2 type and Luminal B (HER2⁻) tumours. We found an association between *PAK1* CN increase, and high proliferation, and high histological grade, but not prognosis. Of cases with *PAK1* CN $\geq$ 6, 30% also had *CCND1* CN $\geq$ 6.

(GDPR), National health research legislation and the conditions for approval by the Regional Committee for Medical and Health Research Ethics, Midt-Norge (REK 836/2009), but may be available from the corresponding author on reasonable request and/or the Institutional Research Officer, Department of Clinical and Molecular Medicine, Faculty of Medicine, NTNU at postmottak@mh. ntnu.no.

**Funding:** The research leading to these results received funding from The Liaison Committee between the Central Norway Regional Health Authority and the Norwegian University of Science and Technology (NTNU), The Joint Research Committee between St. Olav's Hospital and the Faculty of Medicine and Health Sciences, NTNU (FFU), and the Department of Clinical and Molecular Medicine, NTNU. The funders had no role in study design, data collection and analysis, decision to publish, or preparation of the manuscript.

**Competing interests:** The authors have no conflicts of interest to declare that are relevant to the content of this article.

## Conclusions

*PAK1* copy number increase is associated with high proliferation and high histological grade, but not with prognosis. *PAK1* CN increase was most frequent in the HER2 type and Luminal B (HER2⁻) subtype. *PAK1* CN increase is associated with CN increase of *CCND1*.

## Introduction

P21-activated kinases (PAK) are a family of serine/threonine protein kinases comprising six isoforms (*PAK1*–6). They are overexpressed in several human tumours, such as breast cancer (BC), colon cancer and lung cancer, and in neurofibromatosis [1]. The six PAK isoforms are subdivided in PAK1-3 (group I) and PAK4-6 (group II) [2, 3]. PAKs play a significant role in proliferation, cytoskeletal dynamics, and cell survival [1, 4]. Their roles in these cell processes make them potential therapeutic targets. More is known of the functions of PAK1 and PAK4, than of the other isoforms [5, 6].

*PAK1* is located on chromosome 11 (q13.5-q14.1). Amplification of *PAK1* and high PAK1 protein levels are found in several human cancers, including BC [7–9], and are linked to aggressive tumour types, chemotherapy resistance and poor prognosis [4, 10–14]. In 2000, Mira *et al.* first discovered that *PAK1* had an important role in proliferation in BC cell lines [15]. Since then, *PAK1* has been found to be involved in many stages of the BC process and is known to regulate several signaling pathways. [4, 16–21]. *PAK1* amplification has recently been found to be significantly associated with reduced relapse-free survival of ER-positive BC patients [19]. *PAK1* is localized in the same chromosomal region as *CCND1*, 11q13 [22, 23]. Cyclin D1 (CCND1) has been found to be overexpressed in breast cancer, and *PAK1* is shown to regulate the expression of CCND1 in BC [8, 23].

In this study we aimed to assess *PAK1* gene copy number (CN) in a well-characterized series of primary BCs and their corresponding axillary lymph node metastases. We studied associations between *PAK1* CN and proliferation, molecular subtypes, and prognosis. In addition, we examined associations between CN of *CCND1*, assessed in an earlier study by our group [24], and *PAK1* CN.

## Materials and methods

### Study population

A population-based survey for the early detection of BC was conducted in the county of Nord-Trøndelag, Norway, between 1956 and 1959. The study included 25,727 women born 1886–1928 [25]. These women were followed for BC occurrence, through linkage with data from the Cancer Registry of Norway. During the follow-up years, between 1961 and 2008, 1379 new BCs were registered. Of these, 909 cases were included in the study population and were first reclassified into molecular subtypes in a previous published by our group in 2013 (Table 1) [26]. All patients were followed from time of diagnosis until death or December 31st, 2015.

For the present study, we performed fluorescence *in situ* hybridization (FISH) on tissue specimens from cases mainly diagnosed after 1985 (n = 558). Of these, 46 were excluded due to missing or insufficient tumour tissue (n = 25), or due to unsuccessful FISH (n = 21). Thus, 512 cases were suitable for assessment of *PAK1* and chromosome enumeration probe 11 (CEP11) CN in primary tumours (Fig 1). Of the 512 cases, 172 had lymph node metastases, and tissue from lymph node metastases was available for 143 cases. Cases with unsuccessful

**Table 1. Reclassification of breast cancers into molecular subtypes [26].**

| Molecular subtype | Classified by |
|---|---|
| Luminal A | ER$^+$ and/or PR$^+$, HER2$^-$, Ki-67<15% |
| Luminal B (HER2$^-$) | ER$^+$ and/or PR$^+$, HER2$^-$, Ki-67$\geq$15% |
| Luminal B (HER2$^+$) | ER$^+$ and/or PR$^+$, HER2$^+$ |
| HER2 type | ER$^-$, PR$^-$, HER2$^+$ |
| Basal-like | ER$^-$, PR$^-$, HER2$^-$, CK5$^+$ and/or EGFR$^+$ |
| 5-negative phenotype | ER$^-$, PR$^-$, HER2$^-$, CK5$^-$, EGFR$^-$ |

*ER = Oestrogen receptor, PR = Progesterone receptor, HER2 = Human epidermal growth factor receptor 2, CK5 = Cytokeratin 5, EGFR = Epidermal growth factor receptor 1

FISH (*n* = 9) or insufficient amounts of tumour tissue (*n* = 11) were excluded. Hence, lymph node metastases from 123 cases were included in the analyses.

## Specimen characteristics

The primary tumours were previously reclassified into histological type and grade according to present-day guidelines [26–28]. Tissue microarray (TMA) blocks were made using the TissueArrayer Minicore with TMA Designer2 software (Alphelys). Three 1-mm in diameter

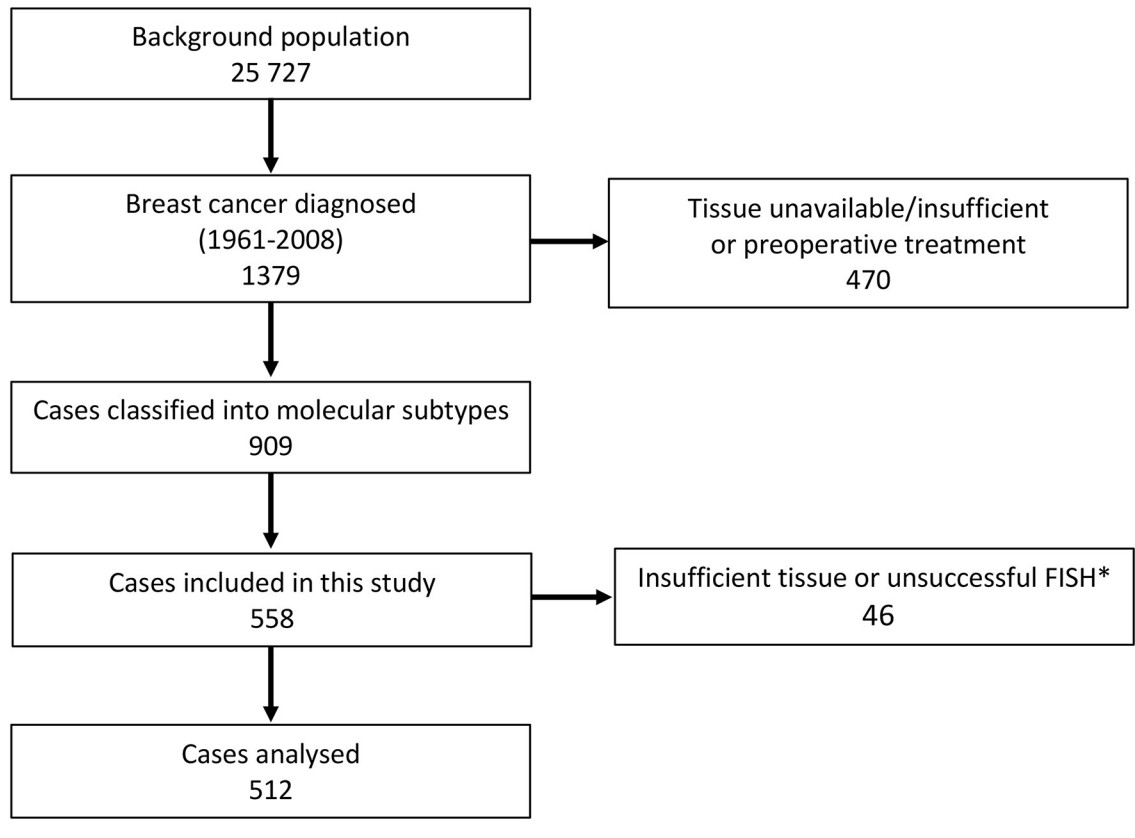

*FISH = fluorescence in situ hybridization

**Fig 1. Overview of study population and cases included in this study.**

**Table 2. Sources and dilutions of primary antibodies used for molecular subtyping [26].**

| Antibody | Clone | Manufacturer | Concentration of antibody | Dilution |
|---|---|---|---|---|
| ER | SP1 | Cell Marque | 33 mg/ml | 1:100 |
| PR | 16 | Novocastra | 360 mg/l | 1:400 |
| HER2 | CB11 | Novocastra | 3.9 g/l | 1:640 |
| Ki-67 | MIB1 | Dako | 35 mg/l | 1:100 |
| CK5 | XM26 | Novocastra | 50 mg/l | 1:100 |
| EGFR | 2-18C9 | Dako | Ready to use | No dilution |

tissue cylinders were extracted from the periphery of the primary tumour, and from lymph node metastases and transferred to TMA recipient blocks. Using sections from the TMAs, primary tumours were then reclassified into molecular subtypes using immunohistochemistry (IHC) and chromogenic *in situ* hybridization (CISH) as previously described (Table 1). Briefly, Oestrogen Receptor (ER), Progesterone Receptor (PR), the proliferation marker Ki-67, Cytokeratin 5 (CK5) and Epidermal Growth Factor Receptor 1 (EGFR) were assessed using IHC, and Human Epidermal Growth Factor Receptor 2 (HER2) was assessed using both CISH and IHC [26] (Table 2). In a previous study of *CCND1* CN, FISH was used to target *CCND1* and CEP11, using Dako Histology FISH Accessory Kit K 579911 probes for *CCND1* (3 μL, Empire Genomics) and CEP11 (1 μL, Abbott/VYSIS) [24].

## Fluorescence in situ hybridization

For the present study of *PAK1* and CEP11 CN, FISH was done using DAKO Histology FISH Accessory Kit K 579911 according to the manufacturer's instructions. TMA sections were preheated at 60˚C for 1–2 h, then de-waxed and rehydrated. The slides were then boiled in a microwave oven for 10 min. in pretreatment solution and washed in DAKO wash buffer (2x3min.) after cooling (15 min.), followed by protein digestion in pepsin solution (37˚C, 25 min.). After protein digestion, the slides were washed in DAKO wash buffer (2x3 min.), dehydrated (2 min. in 70%, 85% and 95% ethanol), then air-dried for 15 min. at room temperature.

*PAK1* (3 μL, PAK1-20-RE, SpectrumRed fluorochrome Empire Genomics) and CEP11 (3 μL, CEP11 [D11Z19], SpectrumGreen fluorochrome, VYSIS) probes were mixed with hybridizing buffer (9 μL, Empire Genomics) and applied to TMA slides according to the manufacturer's instructions. Coverslips were then applied to the slides, sealed with DAKO coverslip sealant, and the slides were dried for 20 min. After drying, denaturation was performed at 83˚C for 3 min., followed by hybridization at 37˚C overnight using DAKO hybridizer. Post-hybridization washes were done in 0.4 X SSC/ 0.3% NP-40 stringent wash buffer at 72˚C (2 min.) and 2 X SSC/ 0.1% NP-40 wash buffer at room temperature (1 min.). Slides were then dried at 37˚C for 15 min., DAPI II VYSIS (15 μl, no 06J50-001) was applied. The slides were then coverslipped and stored at −20˚C.

## Scoring and reporting

A fluorescence microscope (Nikon Eclipse 90i) was used for counting *PAK1* and CEP11 CN. For each case, all available tissue spots were examined and the number of fluorescent signals for *PAK1* and CEP11 were counted in 20 well-preserved, non-overlapping tumour cell nuclei. Mean *PAK1* and CEP11 CNs was calculated for tumours and lymph node metastases and were first categorized as <4 and ≥4. In addition, to distinguish between low-level CN gain and high-level gain or gene amplification, we also subdivided CN into three categories: <4; ≥4<6; and ≥6 according to guidelines for categorizing *HER2* CNs [29], a strategy which has been

used in previous studies of other genes by our group [24, 30–32]. The Reporting Recommendations for Tumor Marker Prognostic Studies (REMARK) were followed [33].

### Statistical analyses

Pearson's chi square test was used to compare tumour characteristics across categories of *PAK1* mean CN. Cumulative incidence of death from breast cancer was estimated, and Gray's test was used to compare equality between cumulative incidence curves. Cox proportional hazard analyses were used to estimate hazard ratios (HR) of breast cancer death with 95% confidence intervals (CI). The analyses were adjusted for age ($\leq$ 49, 50–59, 60–64, 65–69, 70–74, $\geq$ 75), stage (I–IV), histological grade (1–3), and Ki67 status ($</\geq$ 15%). Adjustments were made for each variable separately, and for age, grade, and stage combined. No clear violations of proportionality were observed in log minus-log plots. All statistical tests were two-sided and statistical significance was assessed at the 5% level. We used Stata 16 (Stata corp., College station, TX, USA) in the statistical analyses.

### Ethics statement

This study was granted approval including dispensation from the general requirement of informed consent, by the Regional Committee for Medical and Health Research Ethics, Midt-Norge (REK 836/2009). All methods were carried out in accordance with relevant guidelines and regulations (The Declaration of Helsinki and national regulations (ACT 2008-06-20 no. 44: Act on medical and health research (the Health Research Act)).

## Results

Patient and tumour characteristics for the 512 patients included in the present study are given in Table 3. The mean age at diagnosis was 75.4 years (range 41–96) and the mean follow-up after diagnosis was 9.1 years (SD = 7.2). At end of follow-up, 35.4% of patients had died from BC and 54.3% had died from other causes.

### *PAK1* and CEP11 copy number, and histological grade and proliferation

*PAK1* CN $\geq$4 was found in 48 (9.4%) tumours (Table 3, Fig 2). Of these, 26 (5.1%) cases had mean CN $\geq$4<6, and 22 (4.3%) had mean CN $\geq$6. While 147/464 (31.7%) cases with CN <4 were grade 3, 22/48 (45.8%) cases with CN $\geq$4 were grade 3 (p = 0.037). We found no significant associations between *PAK1* CN increase and high histological grade using three categories of mean *PAK1* CN (Table 3).

 *PAK1* CN $\geq$4 was associated with high Ki-67 ($\geq$15%). Of cases with *PAK1* CN <4, 178/464 (38.4%) had Ki-67 $\geq$15%, compared to 26/48 (54.2%) among those with *PAK1* CN $\geq$4 (p = 0.033). No association between *PAK1* CN increase and Ki-67 status was found when *PAK1* CN was subdivided into three categories. The median mitotic count was higher in cases with mean *PAK1* CN $\geq$4, compared to cases with mean CN <4 (8 mitoses/10 high power fields [HPF] and 5 mitoses/10 HPF, respectively). The proportion of cases with mitotic counts in the upper quartile was also higher for cases with mean *PAK1* CN $\geq$4, compared to those with mean CN <4 (106/464 [22.8%] and 14/48 (29.2%), respectively (p = 0.162)) (Table 3). Only seven cases showed CEP11 CN increase. Five of these were in cases with *PAK*1 CN <4. Of the 26 cases with *PAK1* CN $\geq$4<6, only two were accompanied by CEP11 CN increase ($\geq$4<6). Of the 22 cases with *PAK1* CN $\geq$6, none had concurrent CN increase of CEP11.

**Table 3. Patient and tumour characteristics according to *PAK1* copy number.**

| | Total study population | Mean *PAK1* copy number, three categories | | | | Mean *PAK1* copy number, two categories | | |
|---|---|---|---|---|---|---|---|---|
| | | <4 | ≥4 to <6 | ≥6 | p value ($\chi^2$) | <4 | ≥4 | p value ($\chi^2$) |
| N (%) | 512 | 464 (90.6) | 26 (5.1) | 22 (4.3) | | 464 (90.6) | 48 (9.4) | |
| Mean age at diagnosis, years (SD) | 75.4(41–96) (8.2) | 75.5 (8.1) | 75.2 (7.3) | 74.3 (10.0) | | 75.5 (8.1) | 74.8 (8.6) | |
| Mean follow-up, years (SD) | 9.1 (7.2) | 9.0 (7.0) | 9.6 (6.5) | 9.0 (7.5) | | 9.0 (7.0) | 9.3 (6.9) | |
| Deaths from breast cancer (%) | 181 (35.4) | 161 (34.7) | 9 (34.6) | 11 (50.0) | | 161 (34.7) | 20 (41.7) | |
| Deaths from other causes (%) | 278 (54.3) | 255 (55.0) | 15 (57.7) | 8 (36.4) | | 255 (55.0) | 23 (47.9) | |
| **Histological grade (%)** | | | | | | | | |
| I | 56 (10.9) | 55 (11.9) | 0 (0) | 1 (4.6) | 0.082 | 55 (11.9) | 1(2.1) | 0.037 |
| II | 287 (56.1) | 262 (56.5) | 12 (46.2) | 13 (59.1) | | 262 (56.5) | 25 (52.1) | |
| III | 169 (33.0) | 147 (31.7) | 14 (53.9) | 8 (36.4) | | 147 (31.7) | 22 (45.8) | |
| **Lymph node metastasis (%)** | | | | | | | | |
| Yes | 172 (33.6) | 153 (33.0) | 13 (50.0) | 6 (27.3) | 0.272 | 153 (33.0) | 19 (39.6) | 0.360 |
| No | 228 (44.5) | 209 (45.0) | 9 (34.6) | 10 (45.5) | | 209 (45.0) | 19 (39.6) | |
| Unknown histology | 112 (21.9) | 102 (22.0) | 4 (15.4) | 6 (27.3) | | 102 (22.0) | 10 (20.8) | |
| **Tumor size (%)** | | | | | | | | |
| ≤2 cm | 245 (47.9) | 217 (46.8) | 16 (61.5) | 12 (54.6) | 0.327 | 217 (46.8) | 28 (58.3) | 0.516 |
| >2 cm, ≤ 5 cm | 95 (18.6) | 88 (19.0) | 4 (15.4) | 3 (13.6) | | 88 (19.0) | 7 (14.6) | |
| >5 cm | 10 (2.0) | 9 (1.9) | 1 (3.9) | 0 (0) | | 9 (1.9) | 1(2.1) | |
| Uncertain, but >2 cm | 63 (12.3) | 60 (12.9) | 3 (11.5) | 0 (0) | | 60 (12.9) | 3 (6.3) | |
| Uncertain | 99 (19.3) | 90 (19.4) | 2 (7.7) | 7 (31.8) | | 90 (19.4) | 9 (18.8) | |
| **Stage (%)** | | | | | | | | |
| I | 242 (47.3) | 221 (47.6) | 9 (34.6) | 12 (54.6) | 0.027 | 221 (47.6) | 21 (43.8) | 0.117 |
| II | 218 (42.6) | 198 (42.7) | 14 (53.9) | 6 (27.3) | | 198 (42.7) | 20 (41.7) | |
| III | 27 (5.3) | 22 (4.7) | 3 (11.5) | 2 (9.1) | | 22 (4.7) | 5 (10.4) | |
| IV | 23 (4.5) | 22 (4.7) | 0 (0) | 1 (4.6) | | 22 (4.7) | 1 (2.1) | |
| Unknown | 2 (0.4) | 1 (0.2) | 0 (0) | 1 (4.6) | | 1 (0.2) | 1 (2.1) | |
| **Molecular subtype (%)** | | | | | | | | |
| Luminal A | 272 (53.1) | 251 (54.1) | 11 (42.3) | 10 (45.5) | 0.649 | 251 (54.1) | 21 (43.8) | 0.375 |
| Luminal B (HER2⁻) | 121 (23.6) | 105 (22.6) | 8 (30.8) | 8 (36.4) | | 105 (22.6) | 16 (33.3) | |
| Luminal B (HER2⁺) | 42 (8.2) | 39 (8.4) | 1 (3.9) | 2 (9.1) | | 39 (8.4) | 3 (6.3) | |
| HER2 type | 27 (5.3) | 23 (5.0) | 3 (11.5) | 1 (4.6) | | 23 (5.0) | 4 (8.3) | |
| 5NP | 11 (2.2) | 11 (2.4) | 0 (0) | 0 (0) | | 11 (2.4) | 0 (0) | |
| BP | 39 (7.6) | 35 (7.5) | 3 (11.5) | 1 (4.6) | | 35 (7.5) | 4 (8.3) | |
| **Histological type (%)** | | | | | | | | |
| Invasive carcinoma NOS | 353 (69.0) | 318 (68.5) | 19 (73.1) | 16 (72.7) | 0.593 | 318 (68.5) | 35 (69.0) | 0.273 |
| Lobular carcinoma | 66 (12.9) | 61 (13.2) | 2 (7.7) | 3 (13.6) | | 61 (13.2) | 5 (10.4) | |
| Tubular carcinoma | 1 (0.2) | 1 (0.2) | 0 (0) | 0 (0) | | 1 (0.2) | 0 (0) | |
| Mucinous carcinoma | 24 (4.7) | 23 (5.0) | 1 (3.9) | 0 (0) | | 23 (5.0) | 1 (2.1) | |
| Medullary carcinoma | 14 (2.7) | 10 (2.2) | 3 (11.5) | 1 (4.6) | | 10 (2.2) | 4 (8.3) | |
| Papillary carcinoma | 25 (4.9) | 23 (5.0) | 1 (3.9) | 1 (4.6) | | 23 (5.0) | 2 (4.2) | |
| Metaplastic | 8 (1.6) | 8 (1.7) | 0 (0) | 0 (0) | | 8 (1.7) | 0 (0) | |
| Other | 21 (4.1) | 20 (4.3) | 0 (0) | 1 (4.6) | | 20 (4.3) | 1 (2.1) | |
| **Ki67 high/low (%)** | | | | | | | | |
| Ki67 <15% | 308 (60.2) | 286 (61.6) | 12 (46.2) | 10 (45.5) | 0.104 | 286 (61.6) | 22 (45.8) | 0.033 |
| Ki67 ≥15% | 204 (39.8) | 178 (38.4) | 14 (53.9) | 12 (54.6) | | 178 (38.4) | 26 (54.2) | |
| **Mitoses/10 HPF, median (IQR p25, p75)** | 5 (1, 12) | 5 (1, 11) | 9 (3,20) | 6 (2, 12) | | 5 (1, 12) | 8 (2.5, 16.5) | |

(*Continued*)

**Table 3.** (Continued)

| | Total study population | Mean *PAK1* copy number, three categories | | | | Mean *PAK1* copy number, two categories | | |
|---|---|---|---|---|---|---|---|---|
| | | <4 | ≥4 to <6 | ≥6 | p value ($\chi^2$) | <4 | ≥4 | p value ($\chi^2$) |
| **Mitoses/10 HPF, quartiles (%)** | | | | | | | | |
| ≤1 | 136 (26.6) | 128 (27.6) | 6 (23.1) | 2 (9.1) | 0.025 | 128 (27.6) | 8 (16.7) | 0.162 |
| >1 ≤5 | 133 (26.0) | 123 (26.5) | 1 (3.9) | 9 (40.9) | | 123 (26.5) | 10 (20.8) | |
| >5 ≤12 | 123 (24.0) | 107 (23.1) | 9 (34.6) | 7 (31.8) | | 107 (23.1) | 16 (33.3) | |
| >12 | 120 (23.4) | 106 (22.8) | 10 (38.5) | 4 (18.2) | | 106 (22.8) | 14 (29.2) | |

Abbreviations: SD = standard deviation, HER2 = human epidermal growth factor receptor 2, 5NP = 5 negative phenotype, BP = basal phenotype, HPF = high power fields

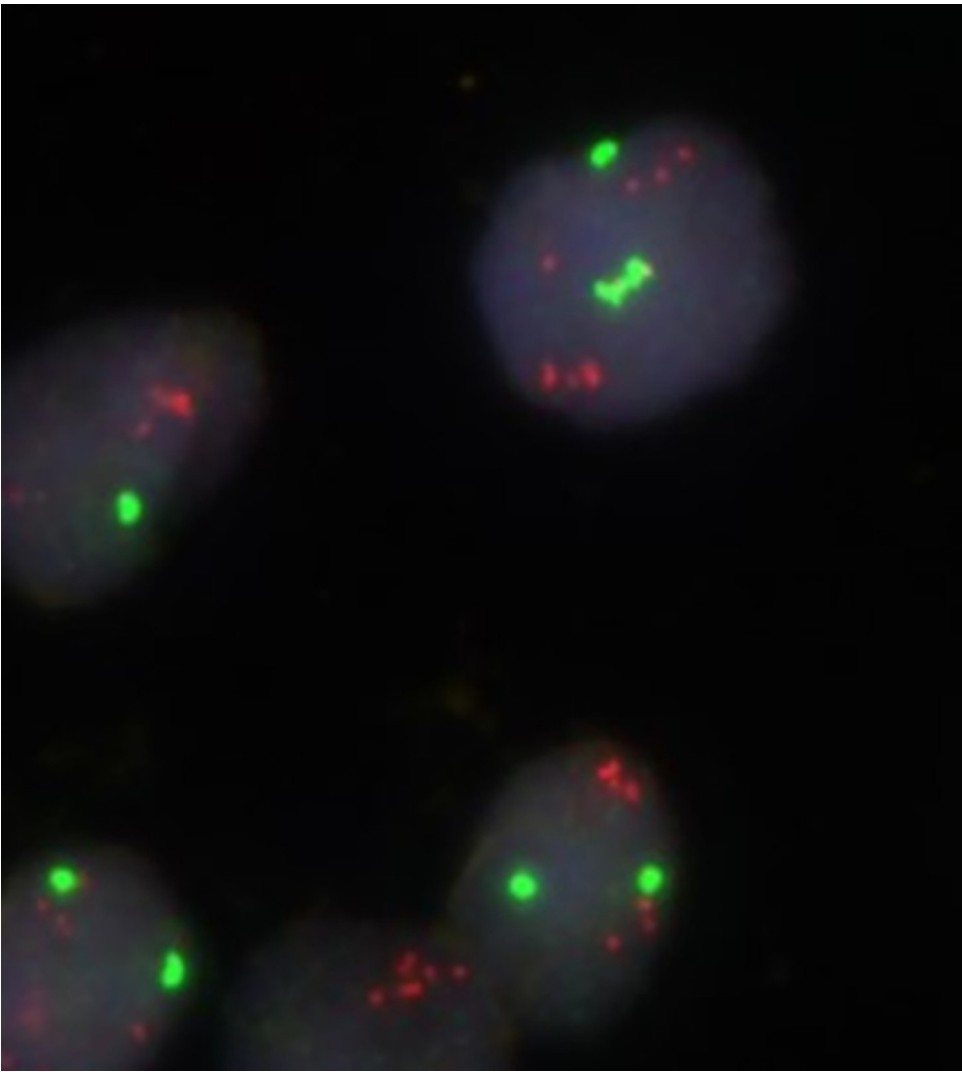

**Fig 2. Fluorescence *in situ* hybridization using probes for CEP11 (fluorochrome SpectrumGreen) and *PAK1* (fluorochrome SpectrumRed).** Fig 2 showing 2–3 copies of CEP11 and 6–8 copies of *PAK1* in each tumour cell nucleus.

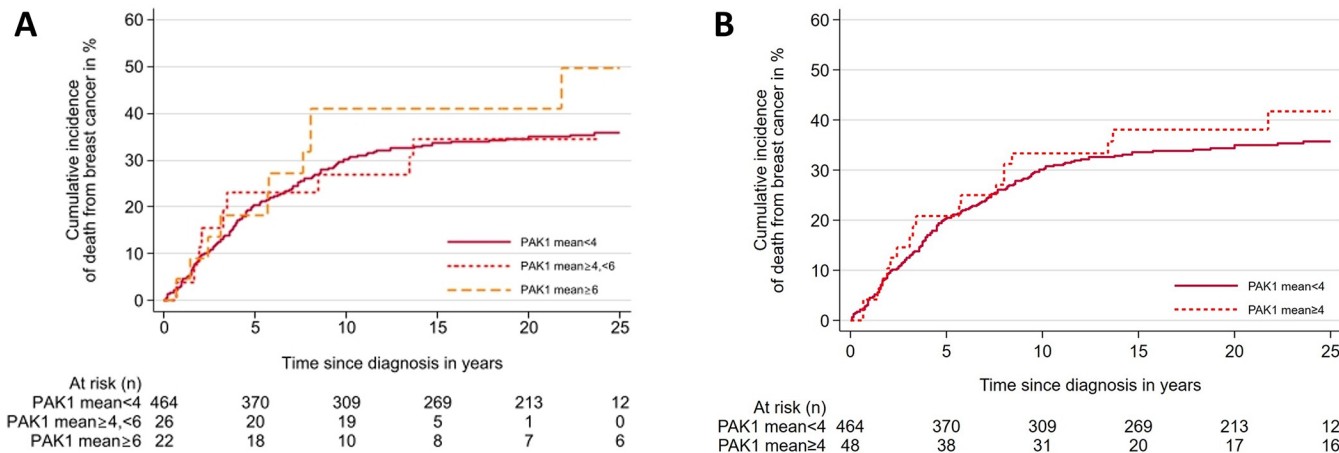

**Fig 3. Cumulative incidence of death from breast cancer according to mean *PAK1* copy number in primary breast cancer tumours.** Cumulative incidence curves show no significant association between *PAK1* copy number and risk of death. A) Mean *PAK1* copy number <4, ≥4<6 and ≥6. p = 0.39. B) Mean *PAK1* copy number <4 and ≥4. p = 0.42.

## *PAK1* copy number and molecular subtypes

Copy number increase of *PAK1* was found in all molecular subtypes, except the 5-negative phenotype (5NP). The highest proportion of cases with *PAK1* CN ≥4 was found in the HER2 type, followed by Luminal B (HER2⁻). Of a total of 27 cases of the HER2 type, four (14.7%) had *PAK1* CN ≥4, one of which (3.7%) had *PAK1* CN ≥6. In Luminal B (HER2⁻), 16/121 (13.2%) had *PAK1* CN ≥4, and of these, 8/121 (6.6%) had *PAK1* CN ≥6. Among Luminal B (HER2⁺) cases, 3/42 (7.1%) showed PAK1 CN ≥4 (Table 3).

## *PAK1* and prognosis

The cumulative risk of death from BC during the first 5 years after diagnosis was 20.3% (95% CI 16.9–24.2) for cases with mean *PAK1* CN <4, 23.1% (95% CI 11.1–44.3) for cases with CN ≥4<6, and 18.2% (95% CI 7.2–41.5) for cases with CN ≥6 (Fig 3, Table 4). During the first 10 years after diagnosis, the cumulative risk of death from BC was 30.1% (95% CI 26.1–34.5) for cases with mean *PAK1* CN <4, 26.9% (95% CI 13.9–48.3) for cases with CN ≥4<6, and 40.9% (95% CI 23.8–63.9) for cases with CN ≥6. In the Cox regression analyses using mean *PAK1*

**Table 4. Absolute and relative risk of death from breast cancer according to mean PAK1 copy number/tumour cell nucleus in primary tumours.**

|  | Mean *PAK1* copy number | | |
| --- | --- | --- | --- |
|  | <4 | ≥4<6 | ≥6 |
| **Cumulative risk after 5 years (%) (95% CI)** | 20.3(16.9–24.2) | 23.1 (11.1–44.3) | 18.2 (7.2–41.5) |
| **Cumulative risk after 10 years (%) (95% CI)** | 30.1 (26.1–34.5) | 26.9 (13.9–48.3) | 40.9 (23.8–63.9) |
| **HR unadjusted (95% CI)** | 1.0 | 0.9 (0.5–1.8) | 1.4 (0.8–2.7) |
| **HR adjusted for age (95% CI)** | 1.0 | 0.9 (0.5–1.8) | 1.5 (0.8–2.7) |
| **HR adjusted for stage (95% CI)** | 1.0 | 0.8 (0.4–1.6) | 1.7 (0.9–3.2) |
| **HR adjusted for grade (95% CI)** | 1.0 | 0.8 (0.4–1.6) | 1.4 (0.8–2.6) |
| **HR adjusted for Ki-67 (95% CI)** | 1.0 | 0.8 (0.4–1.7) | 1.3 (0.7–2.3) |
| **HR adjusted for age, stage, and grade (95% CI)** | 1.0 | 0.8 (0.4–1.5) | 1.7 (0.9–3.2) |

Abbreviations: HR = Hazard ratio, CI = confidence interval

**Table 5. *PAK1* copy number in primary tumours and corresponding axillary lymph node metastases.**

| Mean *PAK1* copy number in lymph node metastases (%) | Mean *PAK1* copy number in primary tumours (%) | | | |
|---|---|---|---|---|
| | <4 | ≥4<6 | ≥6 | Total |
| <4 | 103 (94.5) | 6 (66.7) | 0 | 109 |
| ≥4<6 | 5 (4.6) | 3 (33.3) | 2 (40) | 10 |
| ≥6 | 1 (0.9) | 0 | 3 (60) | 4 |
| Total | 109 | 9 | 5 | 123 |
| Mean *PAK1* copy number in lymph node metastases (%) | Mean *PAK1* copy number in primary tumours (%) | | | |
| | <4 | ≥4 | | Total |
| <4 | 103 (94.5) | 6 (42.9) | | 109 |
| ≥4 | 6 (5.5) | 8 (57.1) | | 14 |
| Total | 109 | 14 | | 123 |

CN <4 as the reference, no significantdifference was observed in the rate of death from breast cancer for cases with *PAK1* CN increase (HR 1.4 [95% CI 0.8–2.7]) for cases with mean *PAK1* copy number ≥6). Fourteen of the 123 cases for which lymph node metastases were available had *PAK1* CN ≥4 in the primary tumour. Of these, 8 also had *PAK1* CN ≥4 in the corresponding lymph node metastasis. Of the five cases with *PAK1* CN ≥6 in the primary tumour, 3 also had *PAK1* CN ≥6 in the corresponding lymph node metastasis (Table 5).

## *PAK1* and *CCND1*

Among the 512 cases included in this study, *CCND1* CN status was available for 504 cases [24]. A total of 84/504 cases showed *CCND1* CN ≥4 and 40 of these had ≥6 copies of *CCND1*/ nucleus (Table 6). Of the 22 patients with *PAK1* CN ≥6, 12 (54.6%) cases also had *CCND1* CN ≥6. Of the 48 cases with *PAK1* CN ≥4, 30 (62.5%) cases also had *CCND1* CN ≥4. However, 54 cases had *CCND1* CN ≥4 without a corresponding increase in *PAK1* CN and 18 cases showed CN increase ≥4 for *PAK1* without CN increase of *CCND1* (Table 6).

We found no significant difference in the cumulative risk of death from BC between cases with CN ≥4 of *PAK1* alone, CN ≥4 *CCND1* alone, and cases with CN ≥4 for both *PAK1* and *CCND1* combined (Fig 4). Similarly, The Cox regression analysis using combined PAK1 CN <4 and CCND CN <4 as the reference value, showed no significant difference in the rate of death from BC between the three groups of patients with copy number increase (Table 7).

**Table 6. *PAK1* and *CCND1* copy numbers in primary tumours.**

| Mean *CCND1* CN | Mean *PAK1* CN in primary tumours (%) | | | | |
|---|---|---|---|---|---|
| | <4 | ≥4<6 | ≥6 | Total | |
| <4 | 402 (88.2) | 11 (42.3) | 7 (31.8) | 420 | p<0.001 |
| ≥4<6 | 31 (6.8) | 10 (38.5) | 3 (13.6) | 44 | |
| ≥6 | 23 (5.0) | 5 (19.2) | 12 (54.6) | 40 | |
| Total | 456 | 26 | 22 | 504 | |
| Mean *CCND1* CN | Mean *PAK1* CN in primary tumours (%) | | | | |
| | <4 | ≥4 | | Total | |
| <4 | 402 (88.2) | 18 (37.5) | | 420 | p<0.001 |
| ≥4 | 54 (11.8) | 30 (62.5) | | 84 | |
| Total | 456 | 48 | | 504 | |

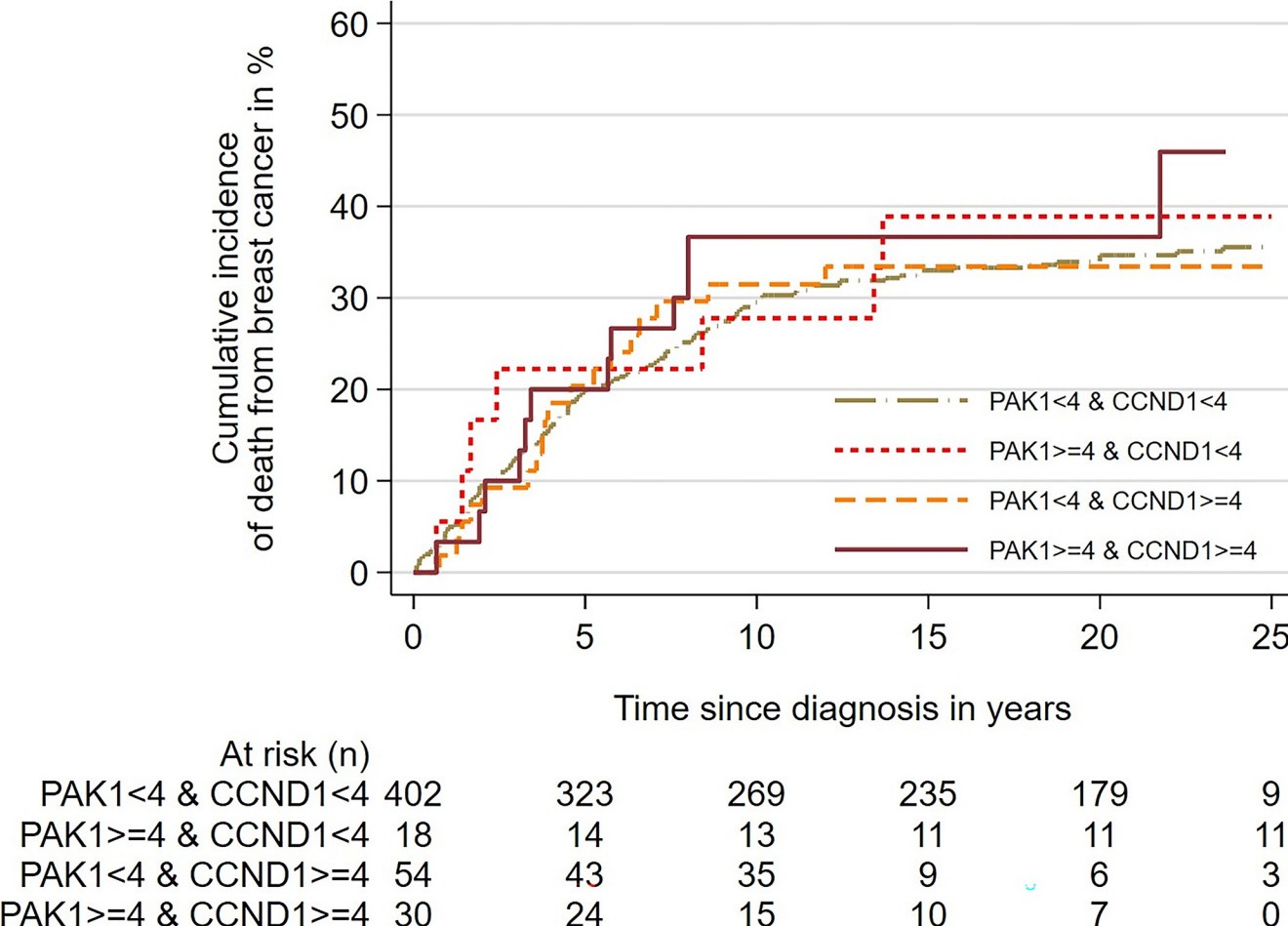

**Fig 4. Cumulative incidence of death from breast cancer according to copy numbers of *PAK1* and *CCND1*, and co-amplification of *PAK1* and *CCND1*.** Cumulative incidence curves show no significant association between *PAK1* copy number, CCND1 copy number, and co-amplification of *PAK1* and *CCND1*, and risk of death. p = 0,81.

## Discussion

In this study of 512 primary BC tumours, we found *PAK1* CN $\geq$4 in 48 (9.4%) cases, of which 22 cases showed high grade CN increase of *PAK1* CN $\geq$6. We found an association between *PAK1* CN $\geq$4, and high Ki-67 ($\geq$15%) and high histological grade. The highest proportion of

**Table 7. Relative risk of death from breast cancer according to copy numbers of *PAK1* and *CCND1*, and co-amplification of *PAK1* and *CCND1*.**

| Copy number of PAK1 and CCND1 | Hazard ratio | | |
|---|---|---|---|
| | HR | CI | p-value |
| **PAK1 CN<4 & CCND1 CN<4 (reference value)** | 1.0 | | 0.872 |
| **PAK1 CN$\geq$4 & CCND1 CN<4** | 1.3 | 0.6–2.6 | |
| **PAK1 CN<4 & CCND1 CN$\geq$4** | 0.9 | 0.6–1.5 | |
| **PAK1 CN$\geq$4& CCND1 CN$\geq$4** | 1.1 | 0.6–2.0 | |

Hazard ratio = HR, Confidence interval = CI

cases with increased CN of *PAK1* (≥4) was found in the HER2 type and Luminal B (HER2⁻) breast cancer subtype. Concurrent CN increase (≥4) of *PAK1* and *CCND1* was observed in 30 cases. Of the 123 cases with available lymph node metastases, only three cases had *PAK1* CN ≥6 in both the primary tumour and the corresponding lymph node metastases.

The cohort of Norwegian BC patients from which the cases of this study are derived is well-described, with mean follow-up of nine years. Since recurrence and death from BC may occur many years after the primary diagnosis, long-term follow-up is important in studies of prognostic markers. While recurrence data was unavailable to us, long-term survival data is complete, enabling us to assess the influence of biomarkers on prognosis. Histological typing and grading of all cases in this cohort were revised by experienced pathologists according to current guidelines. All biomarkers were stained at the same laboratory, and the same antibodies, cut-off levels and algorithm for molecular subtyping were used for all cases in the cohort [26].

In this study we used FISH on TMAs. TMAs provide the opportunity to efficiently study biomarkers in a large number of samples simultaneously under similar laboratory conditions at a relatively low cost. FISH is a method available in most laboratories, contrary to multigene assays. It enables us to assess the morphology of the section and ensure that only invasive tumour cell nuclei are assessed. Despite this, FISH applied to tissue sections may lead to an underestimation of CN compared to analysis of whole nuclei, due to nuclear truncation [34]. This would be of particular importance in cases with low CN increase. Preanalytical conditions will have varied considering that the cases included in the present study were diagnosed over decades. This could have affected the cases suitable for FISH analysis. However, few cases were discarded due to unsuccessful FISH. There are no established guidelines for cut-off levels in the assessment of *PAK1* CN. We chose to follow *HER2* guidelines for categorizing CN, as in previous studies by our group [24, 29–32]. While we also registered CN of CEP 11, we did not calculate the ratio between CNs of *PAK1* and CEP11 as this would have masked the true gene CN increase. Furthermore, we found that CEP11 CN increase was observed in only seven cases, of which only two were accompanied by CN increase of *PAK1*.

Tamoxifen is an established hormonal therapy used in ER positive BC. Five years of tamoxifen therapy nearly halves the risk of BC recurrence among ER positive patients [35]. Phosphorylation of ER by PAK1 may induce tamoxifen-resistance in ER positive tumours and tamoxifen itself may also increase nuclear PAK1 and PAK1 kinase activity [14, 23]. Patients with *PAK1* amplification have reduced benefit from tamoxifen and *PAK1* CN may therefore be a predictor of tamoxifen resistance [23]. Thus, PAK1-inhibitors may be useful in ER positive tumours, to improve the effect of tamoxifen in these cases [36].

Both *PAK1* and *CCND1* encode proteins shown to activate ER [23, 36]. Both are located on 11q13 and are thought to be frequently co-amplified. In this study, of the 504 patients analyzed for both *CCND1* and *PAK1*, 84 cases had CN ≥4 for *CCND1* and 48 with *PAK1* CN ≥4. A total of 30 (62.5%) cases had CN increase of both genes. These results are in accordance with the findings of others [23]. In the present study, co-amplification of *PAK1* and *CCND1* was not associated with prognosis.

The proportion of cases with increased *PAK1* CN in this study was lower compared to the results of previous studies [7, 8]. However, the mean age at diagnosis in our study was 75.4 years, which is high compared to other studies and higher than the mean age for diagnosis of breast cancer in Norway which is 62 years of age [37]. Fumagalli et al found CN increase in 11% of cases in a selected series of ER⁺, metastatic breast cancer cases. In our series of cases, PAK1 CN increase was found among Luminal B HER2⁻ and the HER2 type [38]. High proliferation rate and poor prognosis are found to be associated in BC [39, 40], and the prognostic effect of proliferation has been shown to vary with age, exerting a greater effect on prognosis among younger BC patients [41]. This may, in part, explain the discrepant results compared to

other studies of *PAK1* and further studies including a wider age range are warranted. Furthermore, the choice of method may also have contributed to these results. Tissue microarrays include only small tissue cylinders from the tumour and may not be representative of the whole tumour, particularly in cases with tumour heterogeneity [42, 43]. In the TMAs used in our study, tissue cylinders were extracted from the tumour periphery and are therefore not necessarily representative of other areas of the tumour. However, we considered the tumour periphery to be the region of greatest interest in the tumour given its greater proliferative activity [44] and its proximity to surrounding breast tissue. Furthermore, selecting tissue for TMA form the same region of all tumours contributes to a certain standardization of the material examined in the study.

Despite associations between *PAK1* CN increase and high histological grade and high proliferation, we failed to demonstrate a statistically significant association between increased *PAK1* CN and prognosis. It would be interesting to study prognosis according to *PAK1* CN for each of the molecular subtypes separately. However, in the present study the number of cases in some of the molecular subtypes was too low to warrant further analyses of subgroups. The numbers of cases showing *PAK1* CN increase in primary tumours only, lymph node metastases only, or both were too low to give reliable prognostic information. The frequency of *PAK1* CN change in this study was lower than the expression of established biomarkers, such as ER, PR and HER2 in BC. However, in an era of personalized medicine, its known influence on the effect of tamoxifen in BC makes it an interesting biomarker and potential target for treatment.

## Conclusion

*PAK1* CN increase is found in all molecular subtypes, except the 5-negative phenotype (5NP), and most frequently in the HER2 and Luminal B (HER2⁻) subtypes. It is associated with aggressive tumour characteristics such as high histological grade and high Ki-67 protein expression, but not with prognosis. It is co-amplified with *CCND1* in a proportion of cases. Few cases showed *PAK1* CN increase in both the primary tumour and the corresponding lymph node metastases.

## Acknowledgments

The authors thank the Department of Pathology, St. Olav´s Hospital, Trondheim University Hospital for making the diagnostic archives available for the study, and the Cancer Registry of Norway for supplying the patient data.

## Author Contributions

**Conceptualization:** Anna M. Bofin.

**Formal analysis:** Anette H. Skjervold, Marit Valla, Borgny Ytterhus, Anna M. Bofin.

**Investigation:** Anette H. Skjervold, Marit Valla, Anna M. Bofin.

**Methodology:** Anette H. Skjervold, Borgny Ytterhus, Anna M. Bofin.

**Supervision:** Anna M. Bofin.

**Writing – original draft:** Anette H. Skjervold, Marit Valla, Anna M. Bofin.

**Writing – review & editing:** Anette H. Skjervold, Marit Valla, Borgny Ytterhus, Anna M. Bofin.

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
