## [Decision Letter · Decision Letter 0]

11 Apr 2023

PONE-D-23-02665PAK1 copy number in breast cancer - Associations with proliferation and molecular subtypesPLOS ONE

Dear Dr. Skjervold,

Thank you for submitting your manuscript to PLOS ONE. After careful consideration, we feel that it has merit but does not fully meet PLOS ONE’s publication criteria as it currently stands. Therefore, we invite you to submit a revised version of the manuscript that addresses the points raised during the review process.

We look forward to receiving your revised manuscript.

Kind regards,

Elingarami Sauli, PhD

Academic Editor

PLOS ONE

- https://pubmed.ncbi.nlm.nih.gov/35459982/?

- https://ntnuopen.ntnu.no/ntnu-xmlui/bitstream/handle/11250/2781262/Valla2021_Article_DTX3CopyNumberIncreaseInBreast.pdf?isAllowed=y&sequence=2

- https://link.springer.com/article/10.1007/s10549-020-06035-0

In your revision ensure you cite all your sources (including your own works), and quote or rephrase any duplicated text outside the methods section. Further consideration is dependent on these concerns being addressed.

“The research leading to these results received funding from The Liaison Committee between the Central Norway Regional Health Authority and the Norwegian University of Science and Technology (NTNU), The Joint Research Committee between St. Olav’s Hospital and the Faculty of Medicine and Health Sciences, NTNU (FFU), and the Department of Clinical and Molecular Medicine, NTNU.”

7. Your ethics statement should only appear in the Methods section of your manuscript. If your ethics statement is written in any section besides the Methods, please move it to the Methods section and delete it from any other section. Please ensure that your ethics statement is included in your manuscript, as the ethics statement entered into the online submission form will not be published alongside your manuscript.

8. Please include your tables as part of your main manuscript and remove the individual files. Please note that supplementary tables (should remain/ be uploaded) as separate "supporting information" files

Reviewers' comments:

Reviewer's Responses to Questions

**Comments to the Author**

1. Is the manuscript technically sound, and do the data support the conclusions?

Reviewer #1: Yes

Reviewer #2: Yes

Reviewer #3: Partly

2. Has the statistical analysis been performed appropriately and rigorously? 

Reviewer #1: Yes

Reviewer #2: Yes

Reviewer #3: Yes

3. Have the authors made all data underlying the findings in their manuscript fully available?

Reviewer #1: Yes

Reviewer #2: Yes

Reviewer #3: Yes

4. Is the manuscript presented in an intelligible fashion and written in standard English?

Reviewer #1: Yes

Reviewer #2: Yes

Reviewer #3: No

5. Review Comments to the Author

Reviewer #1: Dear authors,

I read with interest the article which analyzes the associations between PAK1 CN and proliferation status, molecular subtype, and prognosis in addition to the correlation between CNs of PAK1 and CCND1.

I also think that the molecular targets related to the biological aggressiveness of the tumors could constitute a pharmacological target in the personalization of the treatments.

I recommend this article.

The manuscript clear, relevant for the field and presented in a well-structured manner.

The cited references mostly recent publications (within the last 5 years) are relevant.

The manuscript scientifically sound and is the experimental design appropriate to test the hypothesis.

The manuscript’s results reproducible based on the details given in the methods section (to be implemented).

The figures/tables/images/schemes are appropriate and they properly show the data (easy to interpret and understand).

The statistical analysis or data acquired are appropriate.

The conclusions are consistent with the evidence and arguments presented the ethics statements and data availability statements to ensure they are adequate.

The results interpreted appropriately and are significant. All conclusions are justified and supported by the results and the hypotheses carefully identified. The article written in an appropriate way and the data and analyses presented appropriately.

Reviewer #2: The manuscript was well written, the methodology was sound and the results were highlighted systematically.

However, very few issues need address.

1. Line 8 (page 4): Patients were diagnosed, however it was not stated if patients were categorized into treatment options. We assumed molecular classification played role in treatment options and this will be important in the effect of PAK 1 and Tamoxifen as mentioned in the discussion.

2. Table 3 which should highlight the association of PAK 1 CN with Ki67 and Histological grade is missing

3. Kindly check results for line 15 and 16 on page 7 to make sure that comparisons are made with common denominator

4. Line 2 of page 7 described described age demography, with inconsistent gap, while table 3 assumed to highlight this data is conspicuously missing.

5 The discussions on PAK1 CN and CCND1 was scanty.

Reviewer #3: The manuscript titled PAK1 copy number in breast cancer - Associations with proliferation and molecular subtypes is relevant and corroborated knowledge already known.

Abstract

Aims - change to background or introduction. Define all abbreviations first before use.

Method -This was over summarised. More explanation needed.

Result- Add a comment on how the association was found.

Conclusion- Too long. Let it answer the research question

Body of manuscript

Introduction: adequate. Define all abbreviations before use

Material & methods: Adequate and sound. Check the fish protocol and edit appropriately- Make it flow for ease of understanding.

Result: Adequate but need revisions. 'No clear association....' was used several times. It is unclear what it means.

Pg 7 Line 18-25 and 8:1-3 - revise for clarity; HER2- or HER2 superscript - Stick to one

There is repetition of the entire results in the tables- This makes tables/figures redundant. Just mention important findings in-text.

Table 3 missing

Table 5 and 6 confusing. I will suggest the authors stick to the classification stated in the methodology for copy numbers

Discussion: Generally inadequate. Line 4-9: Repetition of result; not necessary. Line 12-19: This was not given attention in the result.

No discussion on comparison of primary tumor with corresponding lymph node metastasis

Why was the TMA taken from tutor periphery?

Conclusion: Fair.

References: Adequate

General: The manuscript needs grammar and spelling check and discussion should be tailored to the results.

6. PLOS authors have the option to publish the peer review history of their article (what does this mean?). If published, this will include your full peer review and any attached files.

Reviewer #1: **Yes: **Paolo Orsaria, MD, PhD;

Reviewer #2: No

Reviewer #3: No

---

## [Author Response · Author response to Decision Letter 0]

5 Jun 2023

The authors would like to thank both the editor and reviewers for their comprehensive and informative assessment of our manuscript. 

Our responses to their comments and suggestions are uploaded as a separate file in this revised submission.

---

## [Editor Report · Decision Letter 1]

8 Jun 2023

PAK1 copy number in breast cancer - Associations with proliferation and molecular subtypes

PONE-D-23-02665R1

Dear Dr. Anette,

We’re pleased to inform you that your manuscript has been judged scientifically suitable for publication and will be formally accepted for publication once it meets all outstanding technical requirements.

Kind regards,

Elingarami Sauli, PhD

Academic Editor

PLOS ONE
---

## [Editor Report · Acceptance letter]

19 Jun 2023

PONE-D-23-02665R1 

*PAK1* copy number in breast cancer - Associations with proliferation and molecular subtypes 

Dear Dr. Skjervold:

I'm pleased to inform you that your manuscript has been deemed suitable for publication in PLOS ONE. Congratulations! Your manuscript is now with our production department. 

Kind regards, 

on behalf of

Dr. Elingarami Sauli 

Academic Editor

PLOS ONE